# Analysis of Exertion-Related Injuries and Fatalities in Laborers in the United States

**DOI:** 10.3390/ijerph20032683

**Published:** 2023-02-02

**Authors:** Margaret C. Morrissey, Zachary Yukio Kerr, Gabrielle J. Brewer, Faton Tishukaj, Douglas J. Casa, Rebecca L. Stearns

**Affiliations:** 1Korey Stringer Institute, Department of Kinesiology, University of Connecticut, Storrs, CT 06269, USA; 2Department of Exercise and Sport Science, University of North Carolina at Chapel Hill, Chapel Hill, NC 27599, USA; 3Department of Physical Education and Sport, University of Prishtina, 10000 Prishtina, Kosovo

**Keywords:** epidemiology, work, heat stress, physical activity

## Abstract

Laborers are particularly vulnerable to exertional injuries and illnesses, as they often engage in heavy physical work for prolonged hours, yet no studies have examined the top causes of catastrophic exertional injuries and fatalities among this population. The purpose of the investigation was to characterize the top causes of exertional injury and fatality within open access, Occupational Safety and Health Administration (OSHA) reportable data. A secondary analysis of OSHA reported injury and fatality data was performed through open access records from OSHA Severe Injury Reports (2015–2022) and OSHA fatality inspection data (2017–2020), respectively. The research team characterized each reported injury and fatality as “exertion-related” or “non-exertion-related. Injury and fatality rates were reported per 100,000 equivalent full-time worker years and included 95% confidence intervals (95% CI). Of 58,648 cases in the OSHA Severe Injury Report database from 2015–2020, 1682 cases (2.9%) were characterized as exertional (0.20 injuries per 100,000 full-time worker years, 95% CI: 0.19, 0.22). Heat-related injuries encompassed 91.9% of the exertional injuries (*n* = 1546). From the 2017–2022 OSHA fatality inspection database, 89 (1.9%) of 4598 fatalities were characterized as exertion-related (fatality rate: 0.0160 per 100,000 full-time equivalent workers, 95% CI: 0.009, 0.0134). The exertion-related fatalities primarily consisted of heat-related cases (87.6%). Exertion-related injuries and fatalities were most reported in Southeast states, in the construction and excavation industry, and among nonunionized workers. As heat stress continues to be recognized as an occupational health and safety hazard, this analysis further highlights the need for targeted interventions or further evaluation of the impact of heat stress on construction and excavation workers, nonunionized workers, and workers in Southeastern states.

## 1. Introduction

Nearly 340 million occupational accidents and 160 million occupational illnesses are reported globally each year, costing the global economy over 1.25 trillion dollars [1]. Consequently, occupational safety and health is a public health priority in the United States. Much research has examined the nature, cause, and characteristics of injuries and fatalities to create effective prevention and risk management strategies [2,3,4,5,6]. However, research initiatives have focused primarily on accident-related data. In contrast, exertion-related injuries and fatalities have received little attention. Exertion-related injuries and fatalities are important to consider, as laborers, workers who engage in physical activity during their job such as farmers, construction workers, and assembly line workers, engaging in heavy physical work for prolonged hours and often in environmental extremes [7]. Exertion-related injuries and fatalities result from bodily dysfunction or disruption of normal physiological processes. Examples include heat-related, cold-related, and cardiac events. Top causes of exertion-related injuries and fatalities have been extensively examined in US high school athletics and the military [8,9,10,11,12,13], which encompass 8 million and 1.3 million people, respectively [14,15]. However, no studies have examined the top causes of catastrophic exertional injuries and death within the much larger population of workers who perform physical exertion in their jobs (approximately 134.5 million in the US) [16].

To evaluate occupational injuries and illnesses, US employers with more than 10 employees in most public sector companies (and some of the public sector dependent on state jurisdictions) are required to keep record and report serious injuries and deaths to the Occupational Safety and Health Administration (OSHA) [17]. The OSHA was created as part of the Occupational Safety and Health Act (1970) to ensure safety of workers by enforcing safety standards, providing training, outreach, education and documenting the frequency, characteristics, and cause of occupation-related injuries and illnesses [17]. Occupational injuries, illnesses, and deaths have dramatically reduced since the Act’s creation, from 38 worker deaths per day in 1970 to 15 worker deaths per day in 2020 [2,16,18]. Still, many worksites remain unsafe and fatalities continue to occur [18]. For example, in 2020, the incidence of total recordable nonfatal injuries in the private industry was 2.7 cases per 100 full-time workers [18] and fatal work injuries 3.4 fatalities per 100,000 full-time workers [18]. These data on injury and fatality rates are reported in OSHA public databases, which are made public to allow researchers to systematically examine the frequency and characteristics of injuries and fatalities. This critical analysis will assist the OSHA, employers, and other safety stakeholders create effective interventions that target vulnerable working populations and worksites.

Therefore, the purpose of this study was to characterize the top causes of exertion-related injury and fatality within the OSHA data. It was hypothesized that heat-related injuries and fatalities would be among the top three causes of injury and death. As a secondary objective, injury and fatality accident data were examined to evaluate whether there were seasonal differences (i.e., differences in summer compared to non-summer) in accident occurrence. It was hypothesized that the summer months (June, July, August, September) would have significantly higher accident-related injuries and fatalities than other months of the year.

## 2. Methods

Institutional Review Board approval was not required as this study was a secondary analysis of deidentified data that examined exertion- and accident-related injuries and fatalities previously aggregated by OSHA. The cause of each reported injury or fatality was classified by the OSHA and the research team, then characterized as either “exertional” or “non-exertional.” “Exertional” was defined as an injury or fatality resulting from bodily dysfunction or disruption of normal physiological processes. Occupational Injury and Illness Classification System codes were reviewed to determine which sections were characterized as exertion-related (Appendix A). All other data were characterized as non-exertional and divided further into two categories: (1) accident-related and (2) non-accident-related. Accident data were defined as injuries or fatalities that did not represent an “intent to harm” (i.e., gunshot, violence). All “intent to harm” cases were characterized as “non-exertional, non-accident related.” Members of the research team (M.M., G.B.) used these inclusion criteria to select cases that were to be considered for analysis.

### 2.1. Data Sources and Extraction for Exertion-Related Injuries and Fatalities

Severe injury data were identified through open-access records from OSHA Severe Injury Reports (www.osha.gov/severeinjury accessed on 15 November 2021) [19]. The Severe Injury Reports is a database that consists of in-patient hospitalization data. The Occupational Injury and Illness Classification System and corresponding codes were utilized to obtain exertion-related severe injury data [20]. Injury data from 2015–2020 were utilized to summarize current leading causes of exertional injuries. The following columns were extracted from the Severe Injury Reports database: employer, event date, city, state, zip code, nature title, case narrative, and event title.

Fatality data were identified through open access records from OSHA fatality inspection data (www.osha.gov/fatalities accessed on 15 November 2021) [21]. Keywords describing exertion-related injuries within the severe injury database were selected a priori (Appendix A) and entered individually in the fatality inspection database for fatality record retrieval. Data from 2017–2020 for fatality data were utilized. The 2017–2020 time frame was chosen as all other OSHA fatality data were archived and did not include descriptive information on each corresponding fatality. We extracted the following information for each OSHA fatality report: employer, event date, city, state, zip code, brief description of event, event title, union status, and occupation. All fatality reports were extracted individually and combined into a new dataset.

Once fatality and severe injury data were extracted from the databases, each record was screened independently by two coders and allocated to groups (“exertional” and “non-exertional”) based on the cause using the case narrative, event title, and Occupational Injury and Illness Classification System code. All disagreements between coders were resolved through discussion, and a third individual was consulted as a “tiebreaker.” The occupation of each injury and fatality report was characterized by the Labor of Bureau Statistics criteria. For example, postal workers were characterized as “Personal Service.” Business size data were reported only for heat-related injuries and fatalities and were collected through accessing publicly available websites related to employer information (i.e., employer websites).

### 2.2. Data Sources and Extraction for Accident Fatalities

A subset of fatality and severe injury data [19,21] that were identified as non-exertional and accident-related were examined in relation to time of year. Time of year was characterized based on month of the year that the event (injury or fatality) occurred and extracted from event date data.

### 2.3. Statistical Analysis

Data were extracted, managed, and analyzed using Microsoft Excel (Microsoft, Redmond, WA, USA). Data are presented as counts with corresponding rate calculations. Injury and fatality data were utilized as the numerator to assess injury and fatality rates. The injury and fatality data used number of hours worked and corresponding equivalent full-time workers reported from the Bureau of Labor Statistics [22] as the denominator for the estimates. In the most recent available data, there were 134,950,000 estimated full-time workers (269,900,000 total hours worked) within working populations that perform physical labor as part of their occupation. All data are reported per 100,000 equivalent full-time worker years and as a yearly average using 2019 data reported from OSHA [22]. Injury and fatality rates are reported with 95% confidence intervals (95% CIs calculated from the reported average margin of errors within the dataset [22]. The rates were calculated using the following equation:(1)$$\mathrm{Rate}\;\left(\mathrm{per}\;100,000\;\mathrm{FTE}\right)=\frac{N}{134,950,000}\times \frac{1}{\mathrm{number}\;\mathrm{of}\;\mathrm{years}}\times 100,000$$

Equation (1): Injury or Fatality Rate Calculation.

FTE = full-time equivalent workers; N = the number of fatalities or injuries; 134,950,000 = total equivalent workers in the US performing physical exertion (working 40 h per week, at least 50 weeks per year).

Rates were calculated only for overall exertion-related injuries and illnesses, as other parameters (i.e., region, industry) lacked the depth to accurately quantify these values.

Accident data were evaluated based on seasonal trends to examine whether there were seasonal differences in accident-related injuries and fatalities. Data were divided into two seasons: summer (June–September) and non-summer (October–May). For injuries, that would include 18 summer months and 54 non-summer months and fatalities included 12 summer months and 36 non-summer months. A Mann–Whitney U-test was performed to assess differences in distribution across accident-related injury and fatality data between summer and non-summer seasons. Accident data are reported as medians (MEDs) and interquartile ranges (IQRs). Alpha level was set at *p* = 0.05.

## 3. Results

### 3.1. Exertion-Related Injuries

Of 58,648 cases in the OSHA Severe Injury Report database from 2015 to 2020, 1682 (2.9%) were characterized as exertional (Table 1). Heat-related injuries encompassed 91.9% of the exertional injuries (*n* = 1546). The remaining exertional injuries consisted of: cardiac events (6%, *n* = 101), pulmonary embolism events (0.06%, *n* = 1), anemia events (0.06%, *n* = 1), anaphylactic shock events (1.07%, *n* = 18), and cold-related injuries (0.89%, *n* = 15; Table 2). Exertion-related injury rates and summary statistics are presented in Table 2. The overall exertion-related injury rate was 0.20 injuries per 100,000 full-time equivalent workers (95% CI = 0.19, 0.22).

The most common injuries were heat-related (injury rate: 0.19 injuries per 100,000 full-time equivalent worker years, 95% CI = 0.18, 0.20), in the Southeast region (*n* = 638), and in the construction and excavation industry (*n* = 334). Within the heat-related injuries, 50.8% (*n* = 855) were within businesses that had over 500 employees, 15.1% (*n* = 254) were businesses with 100–500 employees, 24.4% (*n* = 410) were businesses with fewer than 100 employees, and 9.7% (*n* = 163) were unknown.

### 3.2. Exertion-Related Fatalities

In 2017–2020, there were 4598 fatalities reported in the OSHA fatality inspection database. Of the 4598 fatalities reported, 89 (1.9%) were characterized as exertion-related (fatality rate: 0.0160 per 100,000 full-time equivalent workers, 95% CI 0.009, 0.0134; Table 3). The exertion-related fatalities consisted of heat-related (87.6%, *n* = 78), cardiac events (8.9%, *n* = 8), cold-related (2.4%, *n* = 2), and asthmatic events (1.1%, *n* = 1; Table 4, Figure 1A). Fatalities occurred most in the Southeast (*n* = 28), in the construction industry (*n* = 24, Figure 1B), and among workers who were nonunionized (80.8%). Within the exertion-related heat fatalities, 6.4% (*n* = 5) were within businesses that had over 500 employees, 15.4% (*n* = 12) with 100–500 employees, 69.2% (*n* = 54) fewer than 100 employees, and 8.9% (*n* = 7) were unknown.

### 3.3. Accident-Related Injuries and Fatalities

The average annual number of accident-related injuries in 2015–2020 was highest in October and lowest in April (Figure 1A, Table 5). The distributions of accident-related injuries for summer (MED: 628; IQR: 614 to 641) and non-summer (MED: 588; IQR: 577 to 622) did not significantly differ (*p* = 0.154). For accident-related fatalities in 2017–2020, the average annual number was highest in August and lowest in February (Figure 2B, Table 5). Accident-related fatalities in summer (MED: 60; IQR: 60 to 95) were higher than those not in summer (MED: 56; IQR: 50 to 59; *p* = 0.048). Table 5 presents the number accidents resulting in an injury or fatality per month every year.

## 4. Discussion

This secondary analysis examined the top causes of exertion-related injuries and fatalities within OSHA-reported databases. To our knowledge, this is the first investigation to extract data with the intent to quantify and characterize worker injuries and fatalities resulting from bodily dysfunction (i.e., exertion-related). The top cause of exertion-related injury and fatality was heat-related, accounting for 87.6% and 91.9% of the data, respectively. The most common injuries and fatalities were in the Southeast region, construction and excavation industry, and among nonunionized workers (fatalities only). Interestingly, most exertion-related injuries occurred within businesses with more than 500 employees and fatalities most often occurred in businesses with fewer than 100 employees. Within our secondary objective, we examined seasonal trends (summer vs. non-summer) in accident data. Accident-related fatalities that occurred in summer were higher than in non-summer.

Our findings contribute to the existing literature that heat illness is considered a top cause of injury and fatality within the physically active, which includes military [8,10], athletics [12], and now laborers. Although the injury and fatality rates remain small (0.20 and 0.0160 per 100,000 full-time equivalent worker years), the OSHA characterized heat as one of its top priorities by announcing the start of the rule making progress to implement a mandated federal heat stress standard [23]. Considering this announcement, other key safety stakeholder groups (e.g., American Industrial Hygienist Association, American College of Governmental Industrial Hygienists, National Safety Council, and American Society of Safety Professionals) have recognized the dangers of heat stress and created task forces, working groups, and committees to create educational materials, voluntary standards, and more to help spread awareness. Our findings further support these initiatives to take action to identify heat hazards and implement heat stress management strategies.

Heat-related injuries and fatalities in the current investigation appeared to be reported disproportionally across the US, with most data from the Southeast region. The Southeast is considered a rapidly growing region as the population has increased by 61% since 1980, whereas the rest of the US has grown by only 36% [24]. Although the resident population is rising in the Southeast, which may account for increases in reported injuries and fatalities, there may be a geographical variation in reported heat-related injuries and fatalities suggesting Southeast and Southwest regions are more susceptible than the Midwest, West, and Northeast regions. This contradicts the paradigm that seasonal acclimatization, where workers are exposed to heat stress more frequently, experienced by workers in southern states would reduce heat-related mortality (i.e., greater risk in northern US) [25]. Increased heat-related cases in the Southeast may extend beyond physiologic susceptibility and may be attributed to socioeconomic status [26,27,28,29,30]. Low socioeconomic workers are overwhelmingly exposed to occupational heat stress, with the lowest-paid 20% of workers suffering five times as many heat-related injuries as the highest-paid 20% [31,32]. Families in the South are more likely to earn low incomes compared to other regions in the US, and Jung et al. (2021) reported that in a case-crossover study and spatial analysis evaluating heat vulnerability in Florida that counties with higher average income per person had lower risk of heat-related health outcomes [30,33]. For every $1000 decrease in average income per person, there was a 0.006% increase in heat-related illness emergency room visits [30].

Many counties in the US, such as in Southern states, that are classified as having high levels of social vulnerability are those with a high number of crop workers [34]. Within known and reported cases, it is well established that US crop workers are 35 times more likely to die from heat-related injuries than the general US population [2]. These workers are often immigrants, live in extreme poverty, are required to perform consecutive long days of work in the heat, and are paid by the amount of work they perform [31,32,34,35,36]. Although they are characterized as one of the most vulnerable populations to heat-related injury and fatality, data within our current investigation do not reflect this. In the current study, exertion-related injuries and fatalities were most reported in the construction and excavation industry. Like the agriculture industry (i.e., crop workers), construction workers are required to spend prolonged hours in the heat and perform heavy physical exertion [37,38,39]. Construction workers comprise 6% of the total workforce, and yet they account for 36% of all occupational heat-related deaths from 1992–2016 in the US [37]. Gubernot et al. (2014) reviewed 359 heat-related deaths in 2000–2010 and reported that construction workers were five times as likely to die from heat-related events as all workers [3]. Although it is tempting to characterize construction workers are the most vulnerable based on our study’s findings, it is important to note that these data are focused on reported injuries and fatalities. Many workers may avoid reporting injuries to their employers for fear of losing their job, and therefore it is difficult to highlight the top industries at risk due to this consideration. To adequately highlight specific industries at risk for exertion-related injuries and fatalities, new research or secondary analyses or aggregated databases are warranted to further understand characteristics that place specific industries or workers at risk.

The exertion-related fatality data in the current study reported that 80.8% of all fatalities were among nonunionized workers. Labor unions act as an intermediary between their members and the business that employs them [36]. The purpose of organized labor unions is to fight for better wages, benefit standards, access to health care, limits on working hours, and safe working conditions for their members [36,40,41]. Hagedorn et al. (2016) examined 16 binding contracts with employers in the Pacific Northwest and reported that the union contract language advances many of the social determinants of health, which has been shown to promote the health and well-being of workers [36]. Moreover, Tsao et al. (2016) created an ecological model using area-based measures of income in New York City and premature mortality to examine the reduction in premature mortality that could be achieved had the minimum wage been $15 from 2008 to 2012 [42]. The study revealed that a $15 minimum wage could have averted 2800 to 5500 premature deaths [42]. This is important to consider, as union membership focuses on providing far wages, above or at minimum wage, to workers. Despite the protective nature of a labor union, membership has declined dramatically to 11.3% [43]. Therefore, there may be more nonunionized workers, which accounts for greater reported fatalities.

Contrary to the current study and others, Altassan et al. (2018) reported that among 26,462 workers, union workers had a 51% higher risk of reportable injury [40]. The authors speculated that increased reportable injury in union workers may be due to workers in dangerous jobs more likely joining a union and differential reporting of injuries between union and nonunion companies. There may be a greater willingness for union members to report injuries and illnesses than nonunion members, as union members act under the protection of their union and experience no repercussions for reporting [40]. Union status data were not available for exertion injuries in the current study, but these studies suggest a need to examine the influence of unions on injury and health outcomes.

Lastly, our secondary objective examined whether accidents occurred more frequently during the summer months compared to non-summer months. This analysis was performed to identify whether seasonal trends may exist in reported injuries, as the summer months are often associated with high environmental heat, which has been shown to increase the risk of accidents, time off tasks, and productivity [44,45,46,47]. Although our data reflect a higher number of fatalities during the summer months, it is difficult to determine whether this increase is the result of heat stress or other factors, such as increased work demands (i.e., more workers in summer months). Our finding highlights the need to explore what factors are influencing the higher number of reported fatalities and why we only see this trend in accident-related fatality data.

### 4.1. Limitations

Although this study has many strengths, there are limitations that must be addressed. In the United States, there are few sources of occupational health data that can be easily accessed and analyzed. The major sources of occupational health data, which include the Bureau of Labor Statistics (BLS, used in the current dataset), annual survey of occupational injuries and illnesses, and workers’ compensation records have divergent reporting guidelines and requirements [22,48,49,50]. The reported OSHA injury data included in the current investigation were employer-reported and consisted of in-patient hospitalizations. Therefore, emergency room visits that did not result in admission were not captured, and are not considered reportable according to OSHA standards. This reporting methodology may be limited, as Schramm et al. (2021) examined data from emergency department visits during a Northwestern heat wave (Oregon, Washington, Alaska, Idaho) in May and June 2021, and found that 3504 (MED = 7 per day) heat-related emergency room visits occurred [50]. These data were identified by researchers from text on their reason for visit [50]. The initial consultation to evaluate the reason for a patient to seek emergency department services is performed by a medical provider, who likely have background in characterization of exertion-related injuries such as heat illnesses.

However, the OSHA Severe Injury database relies on employer classification of the incident in which the employer is expected to relay any medical diagnoses by categories provide through the OSHA reporting form and self-reported outcomes from the patient to the employer. The injuries are then submitted to the state and regional OSHA office, which are classified at this level by OSHA employees according to the OSHA classification system [19]. It is unknown if a medical provider is responsible for accurately characterizing each of the injuries. Lastly, the OSHA reporting criteria do not require employers to report injuries that are treated by first aid. OSHA includes “using cold therapy” and “drinking fluids for relief from heat stress” within the definition of first aid. Therefore, injuries such as heat exhaustion that occur on site and are treated using body cooling and/or hydration will not be accounted for. These factors are critically important to consider when attempting to quantify heat-related injuries and illnesses that occur due to work or the working environment.

The nature of the included databases and the criteria for OSHA reporting are responsible for these missed opportunities. Moreover, workers who are immigrants or nonunionized may not report injuries for fear of retaliation. Although there are aspects of the reporting system that can be improved, it should be noted that this federal database is comprised of reported data supported by a mandate. There are very few reporting systems that require reporting to elucidate trends in injuries and fatalities. Lastly, the study included categories (exertion-related, non-exertion-related, accident, non-accident) that were created and coded by the study team. To limit bias associated with the creation of these categories, all keywords and category decisions were determined a priori.

### 4.2. Practical Implications and Future Directions

As the current study identifies heat-related cases as the top exertion-related fatalities and injuries, it is imperative to consider that the rise in extreme heat events will exacerbate these incidents. The frequency of heat-related injuries and fatalities continues to rise, as climate change increases the duration and intensity of heat waves [51,52,53]. It is projected that between 2030 and 2052, the mean global temperature will increase by 1.5 °C [54]. By 2100, three-quarters of the world’s population could face deadly heatwaves [55,56]. In addition to OSHA-reported data, Park al. (2021) reported that in California alone, based on workers’ compensation injury data, there were approximately 20,000 heat-related injuries and illnesses per year that were not accounted for [51]. This number extrapolated to all US workers would be approximately 170,000 heat-related injuries and illnesses per year. These estimates and the current data highlight the significant of the problem of heat as a cause of injury and fatality and call for more interventions to reduce risk [32]. Nonunionized workers in construction and excavation who reside in the Southeast should be studied to determine what intrinsic and extrinsic risk factors increase fatality and injury in this population. These results also support the need for developing heat-related prevention measures in this population and to continue to identify other populations at risk.

## 5. Conclusions

Reported exertion-related injuries and fatalities in OSHA databases primarily consisted of heat-related cases (>85%). These cases were primarily located in Southeastern states, the construction and excavation industry, and in nonunionized workers. There were also seasonal trends in number of reported accident-related fatalities, where more fatalities were reported in the summer months than the non-summer months. More research is warranted to explore factors that influence the immense number of exertion-related injuries and fatalities characterized as “heat-related,” and specific interventions must be examined to identify mechanisms to reduce the increased frequency.

## Figures and Tables

**Figure 1 ijerph-20-02683-f001:**
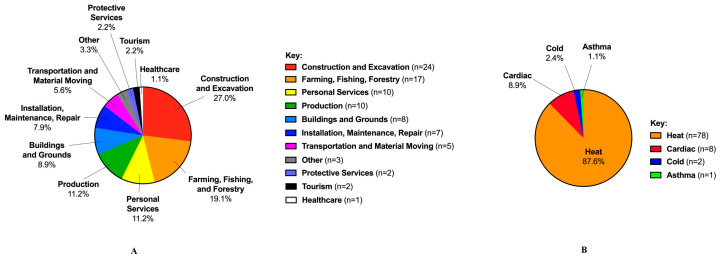
Top causes of exertion-related fatalities across multiple industries. (**A**) Types of causes of exertion-related fatalities; (**B**) industries most affected by ex-ertion-related fatalities.

**Figure 2 ijerph-20-02683-f002:**
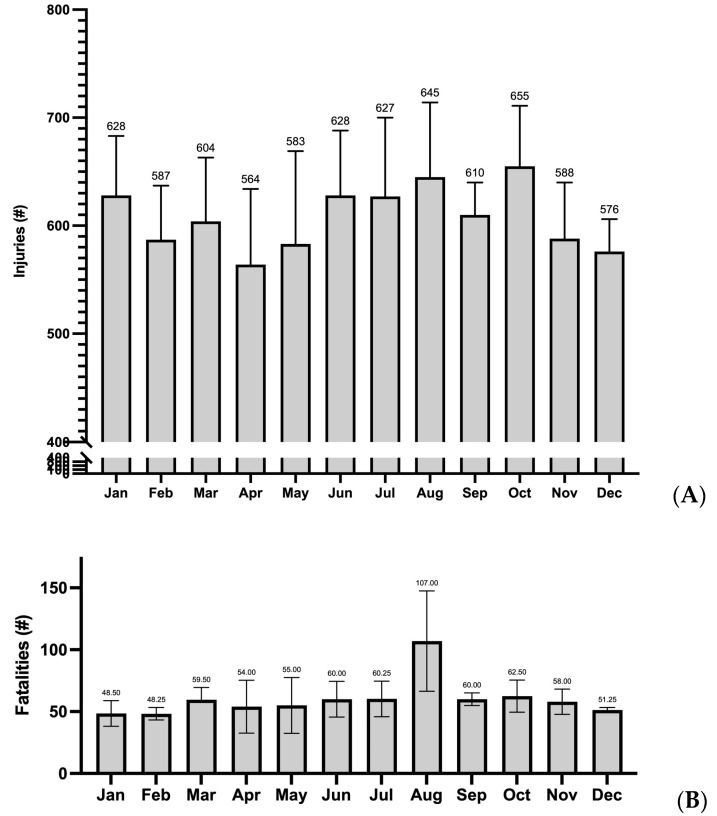
Average number of accident-related severe injury per month (2015–2020). (**A**) Average number of accident-related fatalities per month (2017–2020). (**B**). Note: number above each bar = average number of injuries/fatalities each month.

**Table 1 ijerph-20-02683-t001:** Nature of OSHA reported exertion-related and non-exertion-related injuries in OSHA Severe Injury Report (2015–2020).

Injuries	N	Percent of Overall	Percent of Exertion Injuries
Overall	58,648	100%	-
**Non-Exertion**	56,966	97.1%	-
Accidents *	41,368	70.5%	-
Non-Accidents **	15,696	26.8%	-
**Exertion *****	1682	2.9%	100%
Heat-related Injury	1546	2.6%	91.9%
Cardiac	101	0.2%	6.0%
Anaphylactic Shock	18	0.03%	1.07%
Cold-related Injury	15	0.03%	0.9%
Pulmonary Embolism	1	0.002%	0.06%
Anemia	1	0.002%	0.1%

Note: N = number incidents; * accident data were defined as injuries that were not classified as exertion related or non-accidents; ** non-accidents were defined as injuries that included intent to harm (i.e., violence); *** exertion injury categories created by Severe Injury Report database and classified as “exertion injuries” by research team.

**Table 2 ijerph-20-02683-t002:** Exertion-related injury rates and summary statistics in OSHA Severe Injury Report database (2015–2020).

	Anemia	Cold-Related Injury	Cardiac	Pulmonary Embolism	Anaphylactic Shock	Heat-Related Injury	Total *
**Total**	1	15	101	1	18	1546	1682
Average Yearly Injury Rate ** [95% CI]	0.0001 [0.001, 0.005]	0.002 [0.001, 0.003]	0.012 [0.001, 0.003]	0.0001 [0.001, 0.005]	0.002 [0.001, 0.003]	0.19 [0.18, 0.2]	0.20 [0.19, 0.22]
**US Region**							
Southeast	0	4	15	0	4	615	638
Southwest	1	2	15	0	2	474	494
Midwest	0	4	25	1	5	241	276
Northeast	0	5	36	0	5	168	214
West	0	0	10	0	2	48	60
**Occupation**							
Construction and Extraction	0	3	10	0	0	321	334
Personal Care and Service	1	2	8	0	1	275	287
Production	0	1	7	0	1	256	265
Other	0	3	22	0	0	250	275
Installation, Maintenance, Repair	0	0	11	0	0	155	166
Transportation and Material Moving	0	1	6	1	0	90	98
Buildings and Grounds	0	0	9	0	3	76	88
Food Preparation and Serving	0	3	4	0	1	37	45
Farming, Fishing, Forestry	0	0	0	0	0	41	41
Healthcare	0	1	15	0	11	11	38
Protective Service	0	0	6	0	1	20	27
Tourism	0	1	2	0	0	10	13
Military	0	0	1	0	0	4	5

Note: * Exertion injury categories created by the Severe Injury Report database and classified as “exertion injuries” by research team. ** Reported per 100,000 full-time equivalent workers [(number of injuries/134,950,000) full-time equivalent workers)/6 years].

**Table 3 ijerph-20-02683-t003:** Nature of OSHA-reported exertion-related and non-exertion-related fatalities in OSHA Fatality Inspection Database (2017–2020).

Fatalities	N	Percentage	Percentage of Exertion Fatalities
**Overall**	4598	100%	-
**Non-Exertion**	4509	98.1%	-
Accidents *	2781	60.5%	-
Non-Accidents **	1728	37.5%	-
**Exertion *****	89	1.9%	100%
Heat	78	1.7%	87.6%
Cardiac	8	0.2%	8.9%
Cold	2	0.04%	2.4%
Asthma	1	0.02%	1.1%

Note: N = number of incidents; 95% CI, 95% confidence interval. * Accident data were defined as injuries that were not classified as exertion related or non-accidents; ** non-accidents were defined as injuries that included intent to harm (i.e., violence); *** exertion fatality categories created by fatality inspection database and classified as “exertion fatalities” by research team.

**Table 4 ijerph-20-02683-t004:** Exertion-related fatality rates and summary statistics in OSHA Fatality Inspection Database (2017–2020).

	Heat	Cardiac	Cold	Asthma	Total *
**Total**	78	8	2	1	89
Yearly Average Fatality Rate [95% CI] **	0.0144 [0.0077, 0.011]	0.0014 [0.0004, 0.0019]	0.0003 [0.00005, 0.0009]	0.0001 [0.001, 0.005]	0.0160 [0.008, 0.0134]
**US Region**					
Southeast	26	2	0	0	28
Southwest	23	1	0	0	24
Midwest	13	0	1	1	15
West	11	3	1	0	15
Northeast	5	2	0	0	7
**Occupation**					
Construction and Excavation	24	0	0	0	24
Farming, Fishing, Forestry	15	2	0	0	17
Personal Services	9	1	0	0	10
Production	8	1	0	1	10
Building and Grounds	8	0	0	0	8
Installation, Maintenance, Repair	6	1	0	0	7
Transportation and Material Moving	4	1	0	0	5
Other	2	1	0	0	3
Protective Services	1	0	1	0	2
Tourism	1	0	1	0	2
Health Care	0	1	0	0	1
**Union Status**					
Non-Unionized	63	6	2	1	72
Unionized	15	2	0	0	17

* Exertion fatality categories created by fatality inspection database and classified as “exertion fatalities” by research team; 95% CI, 95% confidence interval; ** reported per 100,000 full-time equivalent workers [(number of fatalities 134,950,000) full-time equivalent workers)/4 years].

**Table 5 ijerph-20-02683-t005:** Number of accident-related injuries and fatalities reported per month per year.

	January	February	March	April	May	June	July	August	September	October	November	December
**Reported Accident-Related Injuries**												
2015	610	509	583	565	523	579	588	589	591	578	526	529
2016	558	571	529	533	515	628	585	701	607	613	572	564
2017	594	592	637	578	609	655	662	663	590	680	573	600
2018	651	629	644	636	720	737	705	732	599	712	668	598
2019	721	652	681	632	634	594	701	645	663	692	601	590
2020	639	574	550	445	499	579	522	544	-	-	-	-
**Mean**	629	588	604	565	583	629	627	646	610	655	588	576
**Reported Accident-Related Fatalities**												
2017	62	49	52	56	69	63	44	70	62	78	57	54
2018	37	45	72	56	70	60	66	74	60	66	44	51
2019	46	44	63	78	59	76	77	138	65	59	67	49
2020	49	55	51	26	22	41	54	146	53	47	64	51
**Mean**	49	48	60	54	55	60	60	107	60	63	58	51

## Data Availability

The data that support the findings of this study are available in the Occupational Safety and Health Administration Fatality Inspection Database and the Occupational Safety and Health Administration Severe Injury Report Database at www.osha.gov/fatalities accessed 15 November 2021 and www.osha.gov/severeinjury accessed 15 November 2021, respectively [19,21].

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
