# Peer review of "Analysis of Exertion-Related Injuries and Fatalities in Laborers in the United States"

_ijerph, 2023, doi:10.3390/ijerph20032683_

Round 1
Reviewer 1 Report
I read this article with interest because not much research addresses this topic from a managerial's perspective. There are numerous medical research papers, some of which are cited by the authors.
As I continued reading, I noticed the following:
- the abstract is quite difficult to understand, it is too long and contains many details that do not belong in this section. This section should contain the main results and their practical implications.
- the references in the text are not properly indicated, so the paper looks like a draft.
- the Literature Review section is included in the Introduction and is insufficiently developed. A more detailed section can be found in the discussion section. I suggest moving an important part of the text from the Discussion section to a Literature Review section that, in this way, can identify some directions for statistical processing.
- the statistical processing is minimal and has the complexity of a term paper. It would be necessary for the authors to go beyond the simple description of a phenomenon using numbers and obtain the results through more complex processing (use of statistical tests, correlations with other variables, possible statistical modeling of the analyzed phenomenon, etc.) to obtain meaningful results.
- the discussion section should refer primarily to the results obtained and refer to other studies only for comparison or to support one's results.
- the Conclusions section should be developed. The section: conclusion, practical implications, and limitations can be created (what has already been written in the discussion section can be moved here).
- the bibliography is not written correctly.
Good luck!
Author Response
We thank the reviewer for their detailed response to improve the submitted manuscript. We have addressed Reviewer #1's comments as follows:
the abstract is quite difficult to understand, it is too long and contains many details that do not belong in this section. This section should contain the main results and their practical implications.
We thank the reviewer for their comments on the abstract section. We have revised the abstract to include only main results and have added the following sentence to address practical implications, “As heat stress continues to be recognized as an occupational health and safety hazard, this analysis has further highlighted the need for targeted interventions or further evaluation of the impact of heat stress on Construction and Excavation workers, non-unionized workers, and workers in Southeastern states”.
- the references in the text are not properly indicated, so the paper looks like a draft.
We have revised the references in the text so that they are properly indicated.
- the Literature Review section is included in the Introduction and is insufficiently developed. A more detailed section can be found in the discussion section. I suggest moving an important part of the text from the Discussion section to a Literature Review section that, in this way, can identify some directions for statistical processing.
We understand the reviewer’s concern regarding the introduction section. The introduction section has been written to focus on exertion-related injuries and fatalities in general, rather than heat-related injuries and fatalities. As heat-related injuries and fatalities were discovered to be the top cause of exertion-related injury and fatality in the current dataset and the characterization of exertion-related events were performed as part of our descriptive analysis, we elected to keep detailed information regarding heat-related injuries and fatalities in the discussion section. We have used published descriptive epidemiological studies of injuries and fatalities (Boden et al. 2013, Gemp et al. 2011, Kerr et al. 2010) as a template when drafting our introduction section.
- the statistical processing is minimal and has the complexity of a term paper. It would be necessary for the authors to go beyond the simple description of a phenomenon using numbers and obtain the results through more complex processing (use of statistical tests, correlations with other variables, possible statistical modeling of the analyzed phenomenon, etc.) to obtain meaningful results.
We understand the reviewer’s concern regarding the statistical analyses used in our manuscript. However, a priori to the conducting of analyses and in the process of developing a research analysis plan, we considered the following: First, the lack of research on the topic with such data warranted analyses more descriptive in nature to “set the stage.” As posited by the van Mechelen “sequence of prevention” framework (Sports Med, 1992), the first step in this process is to identify a baseline, with the hopes that more in-depth analyses can be conducted with data stemming from research methodologies that build upon surveillance such as that from OSHA (Chandran et al., Sports Med, 2020). Second, OSHA data are meant to collect data from all at-risk exposures and can be assumed to be a “census” of the population as opposed to the “sample,” which muddles the use of inferential statistics, which aim to denote the extent to which phenomenon observed within a sample are likely to be representative of the population from which it is derived. For this reason, we elected to keep the use of inferential statistics to a minimum, focusing on future research directions that sample-based research studies could implement. This can be found in the “Practical Implications and Future Directions” section in the discussion.
- the discussion section should refer primarily to the results obtained and refer to other studies only for comparison or to support one's results.
Thank you for your recommendation regarding the discussion section. We have re-written the discussion section to focus on the results obtained and other studies. The two paragraphs pertaining to the nature of the datasets were moved to a Limitations section.
- the Conclusions section should be developed. The section: conclusion, practical implications, and limitations can be created (what has already been written in the discussion section can be moved here).
We have re-ordered the discussion section to include a limitations, practical implications and future directions, and brief conclusion section. We believe the addition of the limitations and practical implications/future directions section support a concise and brief conclusion section in the discussion.
- the bibliography is not written correctly.
We have revised the bibliography so that it is written correctly.
Reviewer 2 Report
Analysis of exertion-related injuries and fatalities in US laborers
Overall comments:
It is true that no studies have examined the top causes of catastrophic injuries and deaths among this population. This study used OSHA's open source data to analyze injury and fatality data. The OSHA Severe Injury Reports with data collected from 1/1/2015 through 5/31/2022 and OSHA fatality reports with data collected from 2017-2020. The data collected from 134,950,000 estimated full-time workers within working populations were extracted, managed, and analyzed using Microsoft Excel, a very simple data processing tool. Only one formula for Injury or Fatality Rate Calculation is used. Rates were only calculated for overall exertion-related including the exertion and overexertion. The exertion data can be filtered in the OSHA excel sheet data.
The study introduced some important conclusions about exertion-related injuries and deaths preeminent in heat-related cases. These cases were primarily located in Southeastern states, the construction and excavation industry, and in non-unionized workers. This is an important study to warn workers how to protect their health, especially in hot weather. The research is very meaningful and has scientific and practical value.
Question:
I have only one question.
As far as I know in the US and European countries, outdoor jobs are mainly implemented in the summer when the weather is warm. Are the conclusions about heat-related injuries and fatalities accurate?
Specific comments:
I have only some minor discussions to contribute to a more complete manuscript.
1. The title of the manuscript is appropriate and concise. However, the abbreviation US should not be used because searching on the internet is difficult and easy to get wrong. Therefore, it should be the United States.
2. The introduction section should describe the terms worker and laborer. What kinds of working populations are mentioned in the manuscript? It may not be necessary to list them all, but it is important to have an overview of working population categories. A brief overview of exertion and overexertion is also needed. The introduction should express when the employee will fall into the situation of exertion and what will happen. A brief overview of the kinds of injuries to workers should also be provided.
3. Part 2. Methods section should be added to the web links to OSHA's data collection sources. I can only download OSHA Severe Injury Reports data. I don’t know how to collect the OSHA fatality reports?
Please describe for more detailed the OSHA data sheet collected, what columns are included, and how many occupations are classified? Where is the source of the number of 134,950,000 estimated full-time workers?
4. The results are made with the following contents: 1/ Analysis of exertion-related Injuries; 2/ Analysis of exertion-related Fatalities and 3/ Analysis of Accident-related Injuries and Fatalities. The study only took into account exertion-related injuries and fatalities. If the research could use the OSHA data to compare with other related injuries and fatalities, it would be more interesting. Outdoor jobs are mainly deployed in the summer when the weather is warm, as I mentioned in the question section, so the possibility of working accidents is higher, but there is no mention of this issue in the results. So, please consider the last conclusion sentence in page 7.
The figures should be re-presented in color to make them more beautiful and attractive.
Conclusions:
This is a study of scientific and practical significance, the research results can help employers and employees take measures to reduce accidents. However, the study is quite simple, only analyzing data as published by OSHA. After removal, only 1682 exertion-related injury cases and 89 exertion-related fatality cases remained, which is not much data. The obtained results are just some of the author's observations and analysis. Therefore, in order to upgrade for more elaborate research, more research results should be obtained.
Author Response
We thank the reviewer for their detailed response to improve our submitted manuscript. We have address Reviewer 2's comments as follows:
As far as I know in the US and European countries, outdoor jobs are mainly implemented in the summer when the weather is warm. Are the conclusions about heat-related injuries and fatalities accurate?
We have included both outdoor and indoor work within the injury and fatality data and all 12 months of the year. The classification of each exertion-related injury were selected from pre-determined categories among The Occupational Injury and Illness Classification System Codes. The Occupational Injury and Illness Classification System are listed in Table S1. For fatality data, the Occupational Injury and Illness Classification System codes and the brief descriptor of the event were used to determine the classified of each exertion-related case. We have clarified these statements in the methods section of the manuscript.
Specific comments:
I have only some minor discussions to contribute to a more complete manuscript.
- The title of the manuscript is appropriate and concise. However, the abbreviation US should not be used because searching on the internet is difficult and easy to get wrong. Therefore, it should be the United States.
We have revised the title to, “Analysis of exertion-related injuries and fatalities in laborers in the United States.
- The introduction section should describe the terms worker and laborer. What kinds of working populations are mentioned in the manuscript? It may not be necessary to list them all, but it is important to have an overview of working population categories. A brief overview of exertion and overexertion is also needed. The introduction should express when the employee will fall into the situation of exertion and what will happen. A brief overview of the kinds of injuries to workers should also be provided.
We thank the reviewer for their insightful comment regarding the terminology , “laborer” and descriptors of exertion-related injuries and illnesses. We have revised the following section of the introduction to address this, “Exertion-related injuries and fatalities are important to consider as laborers, workers who engage in physical activity during their job such as farmers, construction workers, assembly line workers, engage in heavy physical work for prolonged hours and often in environmental extremes[7]. Exertion-related injuries and fatalities result from bodily dysfunction or disruption of normal physiological processes. Examples include heat-related, cold-related, and cardiac events.”
- Part 2. Methods section should be added to the web links to OSHA's data collection sources. I can only download OSHA Severe Injury Reports data. I don’t know how to collect the OSHA fatality reports?
We have included the weblinks to the data sources in the methods section.
Please describe for more detailed the OSHA data sheet collected, what columns are included, and how many occupations are classified? Where is the source of the number of
134,950,000 estimated full-time workers?
In the methods section, we have included that for the severe injury reports database, employer, event date, city, state, zip code, nature title, case narrative, and event title were utilized (first paragraph in 2.1). We have revised the sentence to, “The following columns were extracted from the Severe injury reports database: employer, event date, city, state, zip code, nature title, case narrative, and event title.”.
Similarily for the fatality database, we have indicated that we utilized the following categories: employer, event date, city, state, zip code, brief description of event, event title, union status, occupation. We have revised the sentences to, “We extracted the following information for each OSHA fatality report: employer, event date, city, state, zip code, brief description of event, event title, union status, occupation. All fatality reports were extracted individually and combined into a new dataset.”.
OSHA does not regularly report number of laborers and we used the most recent report for this information. While the volume of at-risk laborers is hard to estimate, the total hours worked in 2019 according to OSHA equated to 134,500,000 full-time workers (269,900,000 total hours worked) citation: https://www.bls.gov/news.release/cfoi.nr0.htm). We indicated that , “ The injury and fatality data used number of hours worked and corresponding equivalent full-time workers [22] as the denominator estimates. There were 134,950,000 estimated full-time workers within working populations that perform physical labor as part of their occupation. All data were reported per 100,000 equivalent full-time worker years and reported as a yearly average using 2019 data reported from OSHA [22]. “ in the statistical analysis section. We have revised the following sentences to clarify, “The injury and fatality data used number of hours worked and corresponding equivalent full-time workers reported from the Bureau of Labor Statistics [22] as the denominator estimates. There were 134,950,000 estimated full-time workers (269,900,000 total hours worked) within working populations that perform physical labor as part of their occupation”.
- The results are made with the following contents: 1/ Analysis of exertion-related Injuries; 2/ Analysis of exertion-related Fatalities and 3/ Analysis of Accident-related Injuries and Fatalities. The study only took into account exertion-related injuries and fatalities. If the research could use the OSHA data to compare with other related injuries and fatalities, it would be more interesting. Outdoor jobs are mainly deployed in the summer when the weather is warm, as I mentioned in the question section, so the possibility of working accidents is higher, but there is no mention of this issue in the results. So, please consider the last conclusion sentence in page 7.
We thank the reviewer for their comment. To address this concern, we had reported non-exertional injuries and fatalities (i.e., accident data) within the manuscript. The objective of the manuscript was to characterize the top causes of exertion-related injuries and fatalities.
As mentioned previously, the first step in this process is to identify a baseline, with the hopes that more in-depth analyses can be conducted with data stemming from research methodologies that build upon surveillance such as that from OSHA. We have included a “practical implications and future directions section” in the discussion of the manuscript.
The figures should be re-presented in color to make them more beautiful and attractive.
We have updated the figures to be re-presented in color.
Conclusions:
This is a study of scientific and practical significance, the research results can help employers and employees take measures to reduce accidents. However, the study is quite simple, only analyzing data as published by OSHA. After removal, only 1682 exertion-related injury cases and 89 exertion-related fatality cases remained, which is not much data. The obtained results are just some of the author's observations and analysis. Therefore, in order to upgrade for more elaborate research, more research results should be obtained.
As indicated in our response to Reviewer #1 regarding more research results and analyses, a priori to the conducting of analyses and in the process of developing a research analysis plan, we considered the following: First, the lack of research on the topic with such data warranted analyses more descriptive in nature to “set the stage.” As posited by the van Mechelen “sequence of prevention” framework (Sports Med, 1992), the first step in this process is to identify a baseline, with the hopes that more in-depth analyses can be conducted with data stemming from research methodologies that build upon surveillance such as that from OSHA (Chandran et al., Sports Med, 2020). For these reasons, we have elected to limit statistical analysis and more reported results. We were also interested in data where medical screening, targeted workplace safety standards, and/or on-site medical care could have likely saved. This is not always the case in accident data.
Round 2
Reviewer 1 Report
I read the article and the author's response. I understand why the authors chose to leave an important theoretical part in the discussion section, although I have a different opinion about the structure of an article.
What I do not understand, however, is why the literature review section is missing. The purpose of the introduction is to outline the research's context, state the research's general objectives, and present the structure of the paper.
The literature review section should not be missing because its purpose is to explore certain aspects and support hypotheses. Even if the authors do not work with hypotheses, the "literature review" section is indispensable.
Statistical data processing remains quite simplistic, but that will be the decision of the scientific editor.
For the rest, the other aspects have been well solved.
Author Response
Thank you for this feedback. These recommendations were interpreted as two fold; first being that the structure of the introduction should be adjusted to include a literature review section, with the second asking for the elements of a literature review to be incorporated.
To address the structure of the introduction, the authors would like to defer to the section editor for specific feedback on if the current form violates the publishing format for IJERPH.
To address the second item in regards to aspects that support our goals/hypotheses we have pulled the page numbers and quotes from the current manuscript that address these concerns. Feedback specific to how we can improve or clarify these components would be welcomed.
General context (Pg. 2): “ Examples include heat-related, cold-related, and cardiac events. Top causes of exertion-related injuries and fatalities have been extensively examined in US high school athletics and military[8–13], which encompasses 8 million and 1.3 million people, respectively[14,15]. Yet, no studies have examined the top causes of catastrophic exertional injuries and death within the much larger population of workers who perform physical exertion in their jobs (approximately 134.5 million in the US)”
General objective (Pg. 2): “Therefore, the purpose of this study was to characterize the top causes of exertion-related injury and fatality within the OSHA data. It was hypothesized that heat-related injuries and fatalities would be among the top three causes of injury and death. As a secondary objective, injury and fatality accident data were examined to evaluate whether there were seasonal differences (i.e., differences in summer compared to non-summer) in accident occurrence.”
Structure of the paper (research) (Pg.2): “These data on injury and fatality rates are reported in OSHA public databases, which are made public to allow researchers to systematically examine the frequency and characteristics of injuries and fatalities.”